# Flow Experience Is a Key Factor in the Likelihood of Adolescents’ Problematic TikTok Use: The Moderating Role of Active Parental Mediation

**DOI:** 10.3390/ijerph20032089

**Published:** 2023-01-23

**Authors:** Yao Qin, Alessandro Musetti, Bahiyah Omar

**Affiliations:** 1School of Communication, Universiti Sains Malaysia (USM), Penang 11800, Malaysia; 2Department of Humanities, Social Sciences and Cultural Industries, University of Parma, 43121 Parma, Italy

**Keywords:** TikTok, problematic use, flow, adolescents, parental mediation

## Abstract

TikTok use and overuse have grown rapidly in recent years among adolescents. However, risk factors for problematic TikTok use are still largely unknown. In addition, drawing on the flow theory and parental mediation theory, this study aims to examine how adolescents’ perceptions of enjoyment, concentration, and time distortion affect their problematic TikTok use behavior. Further, we examined the moderating effect of active parental mediation. An online survey in China received responses from a sample of 633 adolescents between the ages of 10 and 19 (males: 51.2%; M_age_ = 15.00; DS = 0.975). Our findings showed that enjoyment was positively associated with concentration and, in turn, with time distortion. We also found significant positive effects of concentration and time distortion on problematic TikTok use. The effect of enjoyment, however, was non-significant, indicating that hedonic mood was not associated with problematic TikTok use. Out of the three moderated relationships examined in this study, only active parental control was found to be a significant moderator for the relationship between concentration and problematic TikTok use. The significant negative moderation result showed that as active parental mediation grows, the impact of adolescents’ concentration on problematic TikTok use is reduced. Future research directions and implications are discussed.

## 1. Introduction

As a result of the rapid development of mobile devices and information technology, users’ consumption habits on the Internet are changing constantly. Some social media users are no longer satisfied with the form of text and pictures; there is a tendency to prefer vivid short videos, especially user-generated content applications such as TikTok, which are characterized by fragmentation, a low threshold, and high transmission characteristics [1]. TikTok is a social media platform that allows users to watch, share, and create short videos [2]. It has been launched in China since 2016, targeting adolescents and young people [3]. The users under the age of 24 account for more than half of TikTok’s total users (61.73%) [4], making TikTok the most popular leisure activity among China’s millennials [5,6].

TikTok enables users to create interactive and recreational videos. Its powerful AI algorithms and content-oriented distribution strategy can present content tailored to users’ preferences; this is the most distinctive feature that sets TikTok apart from its peers [7]. The use of short video applications helps users relax [8,9]. More specifically, the perception of TikTok’s effortlessness, recommendation accuracy, and recommendation serendipity can most likely provide users with a “flow” experience [10], which refers to the optimal experience of doing something completely concentrated, generating an intense sense of enjoyment and satisfaction, and focusing intensely on the task without being aware of the time [11]. In the context of TikTok use, users simply repeat the swipe-up operation, which makes them completely immersed, feel a high level of fun and curiosity, and even lose track of time [12]. However, the constant use of smart digital devices may result in a flow experience that increases the risk of problematic TikTok use [3,13].

Previous studies have shown that excessive smartphone use can trigger the user’s flow experience [14]. As a result, users may be more likely to develop problematic TikTok use [2,13]. This online behavior is associated with a range of physical and psychological problems, including depression, anxiety, and stress, memory loss [3], poor quality sleep, dry and blurred eyes, and social isolation [5,15].

Additionally, given TikTok’s popularity among Chinese adolescents, the problem of excessive TikTok use has raised concerns, especially among parents of teens. Many parents face the task of actively guiding their children’s online activities to increase the possible benefits of media and limit the risks [16]. These parental efforts are called “parental mediation.” In the past, parental mediation was targeted to examine the effects of television (TV) on children and teenagers in media and communication [17], and it recognized the important role of parents in managing and regulating children’s behavior, especially in traditional media such as TV [18]. As digital media gradually become a common phenomenon, parents shift their attention to various online behaviors such as internet use or social media use to avoid its harmful effect on their children [19]. Parental mediation is considered an effective way to help control the problematic behaviors of children [20,21].

In comparison to problematic online gaming and problematic smartphone use, problematic TikTok use is a relatively new phenomenon. It has been gradually noticed by scholars [3,5,12], but it still receives limited attention [2,5,22,23]. Thus, further research is needed to determine the process of users’ gradual involvement in problematic TikTok use and to clarify how parents may alleviate this problematic use behavior among adolescents. Thus, we should ask: how do adolescents gradually become problematically involved in TikTok? And what are the protective factors? Whether the active parental mediation used in traditional media is effective in the context of TikTok needs to be further explored.

Furthermore, based on previous literature, this study employed the flow theory and the parental mediation theory to explore possible risk factors for problematic TikTok use. By examining the influence of experience perception (enjoyment, concentration, and time distortion) generated from TikTok use and borrowing active parental mediation, we explored how to break the negative aspect of flow. This study aimed at providing new insights into the risk and protective factors for problematic TikTok use.

## 2. Theoretical Background and Hypothesis Development

### 2.1. Problematic TikTok Use

Several scholars have used different terms, such as problematic use [12], compulsive excessive [24], pathological [25], or addiction [2] to describe problematic online behaviors [15]. Consistent with most studies [26,27,28], our study uses the term “problematic use.”

Since problematic online behaviors are a spectrum of related but discrete phenomena [29], problematic TikTok use can also be regarded as an independent construct [14]. Referencing other types of social networking services (e.g., Facebook, Instagram, Weibo, etc.), this study defines problematic TikTok use as the uncontrolled and obsessive use of TikTok, which may have negative physical or psychosocial consequences [30,31]. As a progressive form of overuse, the problematic use behavior is configured as loss of control, withdrawal, an inability to reduce use, and negative consequences [12,32].

In addition, given the potentially detrimental consequences of problematic online behaviors, scholars have started to explore their main risk factors. Previous studies have found that users may use social media for social interaction, entertainment, and self-presentation, but excessive use leads to problematic use behaviors, showing symptoms such as loss of control, withdrawal, and relapse [27]. It may deeply affect people’s mental and physical health, which results in a loss of productivity [33]. Studies have found that individuals’ personality traits [34,35], attachment styles [36,37], attitudes [38], life satisfaction [39], and self-esteem [40] are related to problematic online behaviors. These studies examined problematic social media use at the individual level [34]. Subsequently, some scholars have explored the risk factors from the perspective of technology, arguing that the technological environment (such as the quality of the platform) and the online experience are related to problematic social media use [35]. These studies are discussed from the perspective of users and social media platforms, focusing on the factors that users already generate before the use process, such as personality characteristics. However, the influence of the user’s ongoing experience on problematic use during the use process is far from understood.

Flow experiences can make users feel amused and focused, even lose track of time, and, in some cases, foster problematic online behaviors [31]. In other words, social media satisfy users’ psychological needs through an intrinsic reward mechanism. When users perceive benefits, they might continue to be immersed in the process of using, which creates a closed loop of pleasure and eventually leads to problematic online behaviors [12]. While past research focused on the influence of experiential perception during use and problematic use [12], the present study argues that problematic online behavior does not happen in a vacuum but is gradually formed.

### 2.2. Parental Mediation

Parental mediation refers to the parents providing a reasonable method to guide their child’s behavior properly [41]. Previous studies have focused on how parents can reduce the negative impact of the media on their children [42]. In addition, parental mediation drew on Bandura et al.’s (1977) [43] social learning theory, which focused on the negative effects of media, to explore and evaluate the role of traditional TV media in shaping the role of aggression. They regarded parents’ conscious active mediation, such as how different interpersonal communication strategies can maximize the interests of the media’s influence and reduce the negative impact of media on young people’s cognitive development [44].

Additionally, with the increase in problematic short-form video use among adolescents, many solutions have been proposed. More attention is being paid to the adolescents themselves to ease their problematic media use, such as improving their mindfulness [45], self-esteem [46], metacognitive beliefs [47], and self-control [48]. However, it has been found that this strategy can be ineffective because of the special age of adolescents. They often have a hard time overcoming the temptation of new things and usually lack self-control [49]. The family environment is the most permanent and central in the development of children [50]. Thus, parents play an important role in the occurrence and relief of adolescent Internet addiction [51].

The parental mediation theory suggests that parents can mediate and alleviate the negative influence of media on children’s lives, and they often use different mediation strategies to influence children’s media use behavior [52]. Researchers proposed three different mediation strategies: (1) active mediation, (2) restrictive mediation, and (3) co-viewing [53]. These three dimensions are found in parental mediation relationships among children’s TV viewing [54], video game playing [55], and internet usage behavior [56]. However, as Internet usage has become more common among adolescents, some scholars have highlighted that the existing method needs to be extended to further address parental strategies regarding children’s Internet use behavior [57]. Nikken and Jansz 2014 [58] considered added supervision (i.e., monitoring) and technical safety restrictions as new strategies.

Although parental mediation includes several strategies, this study only examined active parental mediation in the context of TikTok use. It is because, first, the restrictive mediation might have a negative impact, leading adolescents to engage in more risky behaviors [59], such as befriending strangers on social networking sites. Second, mobile phones are relatively personal items, and usually, when using mobile phones, people are alone [60]. Thus, it can be difficult for parents and children to use short-form video apps together. Third, a previous study confirmed that parental supervision or monitoring was ineffective in reducing these problem behaviors. Hefner et al., 2019 [61] found that parental monitoring was inefficient in reducing the adverse effects of problematic phone use in children. Finally, installing filters and monitoring software on electronic devices can be a hard task for parents because it requires an advanced level of computer skills that the average adult does not have [52]. Therefore, we only examined active parental mediation in the context of TikTok use.

### 2.3. Flow Experience and Problematic TikTok Use

The flow experience is a state of intense concentration and immersion in an activity [16]. Flow reflects a person’s psychological need for entertainment and pleasure, and it is a continuous state.

Previous studies applied the concept of flow to online activities [62,63,64]. Scholars considered problematic online behavior to be facilitated by the increasing flow of online activity; therefore, the flow experience was a positive predictor of problematic social media use [12]. However, the relationship between flow experience and user behavior was mainly tested in traditional social media environments; whether the flow can still stimulate adolescents’ problematic online behaviors in the context of TikTok remains unknown.

The social media flow experience represents a multi-dimensional construct including enjoyment, concentration, and time distortion [65]. Enjoyment refers to the individual’s hedonic mood, and concentration is the user’s attention fully focused on the activity. It is the optimal experience generated by the high concentration of a limited stimulus domain and one of the most important representations of flow experience [66], while time distortion refers to a person engaged in an optimal experience, usually experiencing short memory intervals [11]. It has been proposed that individuals may unconsciously fall into problematic use if they pursue and obtain flow experience over and over again [67]. However, the association between the dimensions of flow is unexplored.

#### 2.3.1. Enjoyment, Concentration and Time Distortion

Past research has examined enjoyment as an intrinsic motivation for information system use [68]. It was regarded as a feeling of pleasure and an escape from unpleasant reality, since users may pull out their smartphones to escape the busyness of life and look for something fun [69]. Therefore, it can be considered a typical hedonic motivation for conducting online activities that are expected to cause users to concentrate on the content. As the time for adolescents to watch a video is very short, the accumulation of many short videos may cause them to concentrate on watching the content continuously. This long-term immersion then makes them forget the existence of the environment, resulting in a sense of time distortion. This overall feeling of flow may reduce users’ perception of psychologically unpleasant experiences, such as fatigue. Hence, users lose track of their self-awareness [65], are fully immersed in their ongoing activities, and ignore changes in their surroundings [70]. In this study, we expect that enjoyment will lead to concentration, and concentration will lead to time distortion.

Therefore, we hypothesize:

**H1.** *Enjoyment positively predicts concentration*.

**H2.** *Concentration positively predicts time distortion*.

#### 2.3.2. Enjoyment and Problematic TikTok Use

In the media environment, people can get enough pleasure from the activities they are engaged in. It has been proposed that flow experiences result from repetitive behavior and the desire to maintain positive emotions. This increases the frequency and intensity of media consumption, leading to problematic use [71]. The sense of satisfaction and pleasure derived from a flow experience makes people want to re-experience it over and over again. In addition, engaging in flow-generating activities may come at a great cost, and problematic use behavior is a possible consequence [12]. Given the entertaining nature of TikTok, it seems reasonable that the feeling of enjoyment on TikTok has a connection with adolescents’ problematic TikTok use.

Therefore, we hypothesize:

**H3.** *Enjoyment positively predicts problematic TikTok use*.

#### 2.3.3. Concentration and Problematic TikTok Use

A key factor in flow is the constant concentration on one activity, which can even develop into overuse [72]. Flow as an optimal experience can occur when players perceive a balance between their skill and the challenge within the interaction, accompanied by concentration [13]. Since the flow experience offers users feelings of immersion and pleasure, it is likely for them to generate attachment to the media [13], and this has also been found to have a positive influence on gaming addiction [54,63,73]. In addition, for many adolescents, the merging of TikTok’s multiple functions has aroused strong interest. They are very likely to be absorbed in online activities and ignore their surroundings. Over time, the feeling of concentration on TikTok can trigger problematic use.

Therefore, we hypothesize:

**H4.** *Concentration positively predicts problematic TikTok use*.

#### 2.3.4. Time Distortion and Problematic TikTok Use

An important manifestation of online flow is that the user is completely immersed in the online world and disconnected from the real world. As a result, when people enter regions of time stasis, they become immersed in the virtual world and develop a distorted sense of time [13,74]. In this situation, people’s sense of self is impaired, and they lose their sense of time (i.e., their mental clocks run slowly) [67]. Time distortion is a sign, and the greater the distortion perception, the greater the problematic use.

Additionally, each of TikTok’s videos is short and attractive, and adolescents can easily stick to the activity of watching videos. This time-wasting and problematic use have been proven to be an important negative effect of watching TikTok videos [5,12,75,76]. In other words, users may underestimate the time interval between participating in online activities, which leads to problematic use.

Therefore, we hypothesize:

**H5.** *Time distortion positively predicts problematic TikTok use*.

### 2.4. Active Parental Mediation as Moderator

Adolescents and young adults often use social networks to acquire information, entertain, contact, and express themselves [77]. However, overuse of these platforms can become problematic for some adolescents. In traditional media, parents can use mediation to control their children’s TV use [52]. Additionally, with the evolution of online technology, parents’ concerns have shifted to their children’s online screen use [78]. They are facing a challenge in terms of how to protect their children from the negative effects of online activities [52]. However, whether parental mediation can be applied in the context of social media, especially TikTok, remains to be determined.

Parental mediation involves refraining, co-viewing, active mediation; and restrictive mediation, among them, active mediation has been proven to be likely to alleviate the problematic use of online screens [21]. Active parental mediation refers to the discussion between parents and children about the media’s content and what they watch; this method can mitigate the possibility of adverse consequences, such as aggressive behavior or distorted worldview formation [61]. A previous study has confirmed the relationship between problematic social media use, adolescents, and parents [79]. However, children who receive active parental mediation can experience more positive media use outcomes; in contrast, the problematic behavior of online activities can be aggravated [21].

The family, as the most stable environment for adolescents, has been proven to play a positive role in alleviating problematic use behaviors [80], such as controlling adolescents’ screen use through efforts [81,82,83]. Adolescents can freely follow their interests out of pleasure and curiosity to keep using short video apps. This can lead to constant enjoyment based on intrinsic rewards and time-distorted problematic use behaviors. When adolescents are highly focused on enjoying TikTok and have time distortion perceptions, active parental mediation seems to be an effective measure to reduce the effect.

Therefore, we hypothesize:

**H6.** *Active parental mediation negatively moderates the relationship between enjoyment and problematic TikTok use*.

**H7.** *Active parental mediation negatively moderates the relationship between concentration and problematic TikTok use*.

**H8.** *Active parental mediation negatively moderates the relationship between time distortion and problematic TikTok use*.

## 3. Method

### 3.1. Research Design and Measurements

The model of this study was of the reflective-reflective type, in which the items were interchangeable and the removal of an item did not change the essential nature of the underlying construct [84]. This study aimed to explore the factors that predict adolescents’ problematic TikTok use.

We collected data using purposive sampling from Chinese adolescents, aged 10–19 years old. The World Health Organization defines this age group as adolescents [85]. It is important to note that adolescents comprise the largest portion of TikTok users [4]. In addition, this study used a filter question, “How long do you spend on TikTok in a day?” to gauge problematic TikTok use. The use of the inclusion criteria of age category and time spent on TikTok suggests that the study sample was appropriate for this research.

This study has obtained ethical clearance. The consent to participate in this study was sought from the legal guardians of the respondents prior to the data collection stage. The consent form was attached to the invitation letter to participate in the online survey. The link to the survey was given only after getting the completed consent forms from the legal guardians of the respondents. The statements on the purpose of the study and respondents’ right to withdraw at any point of the survey were clearly stated on the cover page of the online survey. The questionnaire underwent the forward-backward translation method from English to Chinese and then from Chinese to English in order to be applied in the Chinese context. An online survey was administered as it offered multiple benefits such as fast response, low cost, and the ability to handle massive questions and many respondents [86,87].

The data collection involves identifying schoolchildren to participate in the survey. This is because those aged between 10 and 19 are mostly still in school. We randomly selected one primary school, one secondary school, and one high school from Hebei Province, China, using the lottery balloting method. It is important to note that the educational system in Hebei Province follows the national public education system, and hence the targeted sample represents the adolescent population in China to some extent. We contacted the school headmasters via email or the official line. Upon agreement, the online questionnaire link was delivered to them, who later helped circulate it to the students’ social media groups in their respective schools.

In addition, we adopted validated measures to examine the key variables of the study. All variables were measured using a 5-point Likert scale, from “strongly disagree” (1) to “strongly agree” (5). Enjoyment was measured using six items (α = 0.957) adapted from Cao et al., 2020 [35]. We modified the original scale by replacing the words “WeChat” with “TikTok.” For the concentration scale, we adopted 3 items (α = 0.956) from Chen et al.’s (2017) [88] and replaced the words “smartphone” with “TikTok” to suit the context of this study. The time distortion scale of Kim and Ko 2019 [89] was used to assess time distortion in this study. The scale of three items (α = 0.948) was slightly modified by replacing the words “game” with “TikTok.” As for active parental mediation, we used Nikken and Jansz 2014′s [58] active parental mediation scale, which consisted of 9 items (α = 0.940). Finally, we adopted Yu and Fu-min 2005′s [90] Internet Dependence Scale to measure problematic TikTok use. The scale consisted of 19 items (α = 0.900), and the words “online” and “smartphone” were replaced with “TikTok” for all of the statements. The Appendix A presents the measurement of each construct.

### 3.2. Pilot Testing

We ran a pilot study before the actual survey to ensure that the clarity of the instructions and the reliability of the measurement were taken care of. A total of 50 respondents were involved in the pilot study. The sample meets the requirement of 10 percent of respondents for the actual survey [91]. The outcomes showed that Cronbach’s alpha values for all variables were all above 0.80, suggesting a high level of acceptable internal consistency. There was no problem of ambiguity or information overload reported during the pilot testing.

### 3.3. Data Collection

We collected data from January to August 2022, and a total of 735 responses were collected. We found 102 invalid responses because of straight-lining issues. As a result, the remaining 633 valid responses were used for further analysis. This sample size was adequate as it met Green’s (1991) [92] criteria for sample calculation for an unknown population. Table 1 showed that 51.2% of respondents were male and 48.8% were female. The respondents were mainly aged between 10 and 19 years, with most of them aged between 15 and 17 (42.51%).

### 3.4. Common Method Bias (CMV)

As the data of this study was collected from a self-administered report from the same person [93], therefore, CMV was the potential risk that needed to be addressed. We applied the marker variable technique to examine CMV, which was recommended when conducting statistical analysis [94]. In addition, we adopted markers (2 items) with no theoretical relationship to our study [95]. Table 2 showed that the value of R^2^ in problematic TikTok use slightly changed from 0.319 to 0.321, which was less than a 10% change after adding the marker variables to the research model. Table 3 showed there was no significant change when with or without Marker. Therefore, the CMV was not an issue for this study [96].

## 4. Data Analysis and Results

### 4.1. The Measurement Model

The research model was tested in a two-stage approach, involving the examination of the measurement model and structural model, using PLS-SEM analysis, as suggested by scholars [97,98]. The results presented for the variables’ Cronbach’s α values ranged between 0.889 and 0.969, and the composite reliability (CR) values ranged between 0.931 and 0.971. Thus, the model had good internal consistency and reliability. Next, all the outer loadings were larger than 0.6, indicating sufficient indicator reliability [99], and the average variance extracted value (AVE) of each construct was larger than the 0.5 thresholds (ranging from 0.641 to 0.856), which showed satisfactory convergent validity [18]. The results were presented in Table 4 Last, the heterotrait monotrait (HTMT) technique was adopted to test discriminant validity [56]. The following Table 5 indicated that the findings met the critical value (lower than 0.85) [100]. 

### 4.2. Structural Model

After establishing the reliability and validity of the instruments, the path relationships with 1000 bootstrap samples were tested [97]. The results of the structural model are presented in Table 6, Table 7 and Appendix B.

#### 4.2.1. Hypothesis Testing

The results indicated that all the hypotheses were supported except for the direct relationship between enjoyment and problematic TikTok use. Enjoyment was a significant predictor of concentration (β = 0.714, t = 32.676, *p* < 0.01 *), which supported H1. Meanwhile, concentration significantly predicted time distortion (β = 0.693, t = 22.050, *p* < 0.01*), which supported H2. Therefore, based on our findings, the relationship between components of flow (enjoyment, concentration, and time distortion) was sequential: enjoyment was the antecedent of concentration, and time distortion was predicted by concentration.

The positive relationships suggest that enjoyment leads to concentration, and concentration leads to time distortion. Meanwhile, both concentration and time distortion significantly predicted problematic TikTok use (β = 0.305, t = 5.474, *p* < 0.01 *; and β = 0.371, t = 7.239, *p* < 0.01 *, respectively), thus H4 and H5 were supported. However, there was no significant relationship between enjoyment and problematic TikTok use (β = -0.101, t = 2.063, *p* < 0.01 *), therefore H3 was rejected. Thus, we found that feelings of concentration and time distortion were the main antecedents that were positively associated with problematic TikTok use; however, there was no significant effect between enjoyment and problematic TikTok use.

This study examined the role of active parental mediation as a moderator in the relationship between flow experience (i.e., enjoyment, concentration, and time distortion) and problematic TikTok use. We found that active parental mediation significantly interacted with concentration to influence adolescents’ problematic TikTok use (H7: β = −0.145, t = 2.363, *p* < 0.01 *). The moderating results showed that a higher level of problematic TikTok use was associated with higher concentration experiences, and the problematic use level was likely to be severe when adolescents lacked active parental mediation. However, the moderating effects were not found in active parental mediation in the relationship between enjoyment, time distortion, and problematic TikTok use (H6: β = 0.110, t = 2.013, *p* > 0.01; H8: β = 0.068, t = 1.612, *p* > 0.01). Thus, H6 and H8 were rejected. As a result, we have identified the significant moderating role of active parental mediation.

#### 4.2.2. Coefficient of Determination (R^2^) and Predictive Relevance (Q^2^)

The overall quality of the model was evaluated by the coefficient of determination (R^2^) and predictive relevance (Q^2^) [97]. As shown in Table 8, our model had satisfactory explanatory power. The Q^2^ of the problematic TikTok use was significantly different from zero (Q^2^ = 0.194). Overall, approximately 31.9% of the variance in problematic TikTok use was explained by this structural model.

#### 4.2.3. Assessment of Goodness of Fit (GoF)

Apart from the coefficient of determination (R^2^) and predictive relevance (Q^2^), we also ran goodness of fit to show the extent to which the sample data represent the data expected from the actual population. When the GoF value is greater than 0.36, 0.25, and 0.10, it is regarded as high, medium, and small, respectively [101]. Table 9 presented the summary of AVE and R^2^ value. This study obtained a GoF value of 0.574, which was over the cut-off value of 0.36 for the large effect size of R^2^. Therefore, this study concluded that the research model performs well in contrast to baseline values.

## 5. Discussion

In this study, we explored the flow experience dimensions (i.e., enjoyment, concentration, and time distortion) as predictors of Chinese adolescents’ problematic TikTok use and the moderating role of active parental mediation in this association. A structural equation modeling approach with moderation analysis was conducted to test the eight hypotheses drawn from empirical studies.

First, we provided evidence that the flow experience (concentration and time distortion) was positively associated with adolescents’ problematic TikTok use. These findings were consistent with previous research on problematic use [12]. This suggests that TikTok can be a highly engaging online world for adolescents, which might provide them with a sense of immersion and distract them from offline activities.

In addition, we found that concentration was a strong predictor of problematic TikTok use. It could be because adolescents’ attention spans cannot last very long. Hence, the playing time of each video was relatively short—only a few minutes—which is meant to sustain users’ concentration on TikTok. Moreover, the algorithmic recommendation system behind this platform is constantly calculated and rehearsed. The videos that users are interested in are continuously presented on the screen. Further, for adolescents, the easy-to-use format and interesting videos of TikTok can easily attract their interest. Although each video is relatively short, the accumulation of many short videos caused the overall usage time to become very long. When adolescents are exposed to short videos for a long time, they eventually develop problematic TikTok use.

In addition, we found that time distortion significantly affected problematic TikTok use. However, when adolescents are deeply immersed in using TikTok, they might lose their ability to perceive time. They cannot be aware of how long they have been immersed in TikTok and forget about their surroundings, gradually developing problematic TikTok use. In addition, adolescents may lack self-control, which can lead them to be hooked up to TikTok [49]. When too much time is spent on TikTok, they tend to concentrate heavily on the content and lose track of time, which can make them more prone to developing problematic TikTok use.

Further, we found, however, no significant effect of enjoyment on problematic TikTok use. One possible explanation is the formation process of flow. The immersed individuals tend to experience cognitive absorption as they concentrate [64] on the content, which distorts their perception of time. Previous studies found that experiencing flow online was particularly important to users’ subsequent behaviors, such as problematic social media use [12], game disorder [63]. These studies considered flow as a holistic construct but did not account for the cumulative experience of flow. This study asserted that the flow experience accumulates from hedonic feeling to cognitive absorption or from concentration to time distortion. Based on our findings, we argued that experiencing enjoyment was the first stage of flow that could not yet predict the problematic use of TikTok. It was the subsequent stages of flow, involving concentration and time distortion, that led to problematic TikTok use.

In addition, our findings were consistent with past research on the positive influence that active parental mediation has on children’s behavior [21,82,102]. Out of the three moderated relationships tested in this study, we found that active parental mediation negatively moderated the relationship between concentration and problematic TikTok use. As a result, the effect of concentration on problematic TikTok use was reduced when active parental mediation increased. For adolescents who overuse TikTok, active parental mediation would help to reduce their concentration and, hence, alleviate their problematic use of TikTok. Thus, we considered that the discussion on the TikTok short video between adolescents and parents could reduce the probability of problematic use problems. This study asserted that adolescents’ concentration can be reduced by increasing parental mediation, which in turn will alleviate the problematic use of TikTok. Active parental mediation, however, had no interaction effects with enjoyment or time distortion to reduce problematic TikTok use. In other words, the parent’s role became ineffective in mitigating the feelings of enjoyment and time distortion that adolescents experienced when they used TikTok.

### 5.1. Implications

This study has several theoretical and practical implications. First, this study is an initial exploration of the antecedents of adolescents’ problematic TikTok use. Several researchers have conducted studies to explore risk factors for problematic TikTok use, such as stress [12], the features of short videos [30], socio-technical factors, and attachment [2]. However, very few studies have explored the influence of users’ experiences on problematic use of TikTok. Thus, this study enhances our understanding of problematic use behavior by focusing on ongoing experiences during TikTok use and identifies flow experience as a critical factor influencing problematic use behavior. We used flow theory to guide our understanding and found that concentration and time distortion were significant predictors of problematic TikTok use. Our findings reveal the potential impact of flow on problematic TikTok use.

Secondly, this study extends its theoretical contribution by combining the application of flow theory with parental mediation theory. Many studies have applied parental mediation in different contexts, such as mobile phones [82], internet use [91], and digital media [92], and mostly adopted it as a critical factor leading to specific behaviors, such as mobile phone dependency [82], problematic online game use [80], and excessive internet use [103], but have ignored its moderating effect. We treat active parental mediation as a moderator to uncover the extent to which it can reduce problematic TikTok use. This study found that active parental mediation and concentration had significant interaction effects that can help reduce problematic TikTok use. In this study, no interaction effect between active parental control and the other dimensions of flow (i.e., enjoyment and time distortion) was found. Hence, this study asserts the importance of parents’ involvement in addressing adolescents’ problematic TikTok use. Our study found support for the application of the parental mediation theory.

Thirdly, our findings have important practical implications for the role of parent mediation as an intervention strategy to help address problematic TikTok use among adolescents. One of the ways is to divert adolescents’ attention from TikTok to reduce their concentration on the platform, which in turn could reduce problematic TikTok use. In doing so, parents should create an effective discussion space with their teens to discuss TikTok and its addictive features. This is because adolescents may not be aware of the advanced algorithm systems embedded in TikTok that can make them unconsciously and continuously addicted to it. In the process of communicating with their parents, adolescents could identify the shallow entertainment meaning of the content and comprehend the video production’s implication behind the screen, such as its purpose, profits, and process. Parents play an important role in educating their teens about the consequences of problematic use of social media, especially TikTok. Adolescents should be regularly reminded that TikTok is the most addictive medium as compared to other social media platforms.

In our study, we find support for these assertions and confirm that active parental mediation is effective in reducing adolescents’ problematic TikTok use in the Chinese context as well. Education is crucial. To educate adolescents, parents should be knowledgeable of the advantages and disadvantages of TikTok, and competent to guide their children’s behavior and interactions in digital spaces.

### 5.2. Limitations and Future Research

Although the contributions of the study are evident, it still has limitations for future research to investigate. Initially, as the problem of excessive use of TikTok gets worse, it is valuable to further explore the antecedents of this problem. This study mainly uses flow theory to explore how enjoyment, concentration, and time distortion affect adolescents’ problematic TikTok use while leaving space for examining other factors that might influence such problematic online behavior. Future studies can examine other variables (e.g., attachment, childhood traumatic experiences, psychopathology) to provide a comprehensive understanding of this phenomenon. Further, this study only considers active parental mediation in alleviating problematic use behavior; other mediation methods, such as co-viewing and parental control, have not been touched. It is important for future research to test different mediation methods to identify their effectiveness in reducing similar problems. In addition, this is a cross-sectional study. This makes it impossible to test causal relationships. It is important to note that the use of purposive sampling, which is a non-probability sampling technique, in this study may cause bias in the selection of respondents and, hence, limit the generalizability of the findings to the overall population. Future research may address this limitation by getting a more representative sample.

Furthermore, this study was conducted in China. As TikTok is a worldwide online application and other countries, especially in the West, seem to face the same issue (problematic TikTok use), it is valuable for future research to conduct cross-cultural research to understand unique contextual factors that may predict problematic TikTok use. Finally, we used survey data to examine the proposed hypotheses. Although the survey is regarded as an effective way to access respondents’ perceptions and behaviors, the data are usually received in uncontrollable environments. Thus, we suggest future studies explore the use of a between-subject experiment to compare a treatment condition (which exposed participants to active parental mediation) with a control group (in which active parental mediation is absent). Another possible method is using focus group discussions to access a more in-depth understanding of this alarming phenomenon of problematic TikTok use among adolescents.

## 6. Conclusions

TikTok is a great innovation. It has many unique application advantages, such as concise content, an easy-to-use format, and efficient playback, which have attracted a growing number of users worldwide to indulge in it and eventually lead to problematic use. The negative effects are pertinent, especially for adolescents. This calls for research to understand the phenomenon and offer solutions to address the problem. This study responds to this call and discovers that flow experiences (i.e., concentration and time distortion) lead to obsessive use of TikTok and that active parental mediation reduces the concentration effect on problematic TikTok use. Although the relationships were tested in a Chinese context, we believe that our findings could be applied to other contexts as well. This is because problematic TikTok use among adolescents has become a universal phenomenon across the globe. Our results have theoretical and practical significance. However, there is still a long way to go in this direction of research, and we call for future research to extend our current knowledge.

## Figures and Tables

**Table 1 ijerph-20-02089-t001:** Demographic Profile of Respondents (*n* = 633).

Demographics Factors	Frequency	Percentage
Gender		
Male	324	51.2%
Female	309	48.8%
Age		
10–11	106	16.71%
12–14	183	28.90%
15–17	269	42.51%
18–19	75	11.88%

**Table 2 ijerph-20-02089-t002:** Comparison of R^2^ values between baseline model and marker-included model.

Relationships	Without Marker Variable	With Marker Variable
Problematic TikTok Use	0.319	0.321
Concentration	0.509	0.509
Time Distortion	0.481	0.483

**Table 3 ijerph-20-02089-t003:** Comparison of Path coefficient (β) between the baseline model and markerincluded model.

Relationships	Path Coefficient Bate (β)	
	Without Marker	With Marker
Enjoyment -> Concentration	0.714	0.714
Concentration -> Time Distortion	0.693	0.694
Enjoyment -> Problematic TikTok Use	−0.101	−0.099
Concentration -> Problematic TikTok Use	0.305	0.301
Time Distortion -> Problematic TikTok Use	0.371	0.375

**Table 4 ijerph-20-02089-t004:** Results Summary for Reflective Measurement Models.

Constructs	Items	Internal Reliability	Internal Consistency Reliability		Convergent Validity
		Outer Loadings	Cronbach’s Alpha	Composite Reliability	Average Variance Extracted
		>0.60	>0.70	>0.70	>0.50
Enjoyment	FE1	0.869	0.938	0.950	0.760
	FE2	0.884			
	FE3	0.898			
	FE4	0.883			
	FE5	0.844			
	FE6	0.853			
Concentration	FC1	0.923	0.916	0.947	0.856
	FC2	0.927			
	FC3	0.926			
Time Distortion	FTD1	0.875	0.889	0.931	0.818
	FTD2	0.929			
	FTD3	0.909			
Active Parental Mediation	APM1	0.778	0.950	0.956	0.709
	APM2	0.852			
	APM3	0.847			
	APM4	0.852			
	APM5	0.895			
	APM6	0.905			
	APM7	0.885			
	APM8	0.774			
	APM9	0.778			
Problematic TikTok Use	PTU1/PTU11	0.700/0.833	0.969	0.971	0.641
	PTU2/PTU12	0.717/0.784			
	PTU3/PTU13	0.763/0.787			
	PTU4/PTU14	0.782/0.750			
	PTU5/PTU15	0.802/0.811			
	PTU6/PTU16	0.862/0.826			
	PTU7/PTU17	0.874/0.806			
	PTU8/PTU18	0.846/0.743			
	PTU9/PTU19	0.845/0.821			
	PTU10/	0.834			

**Table 5 ijerph-20-02089-t005:** Discriminant Validity: Heterotrait Monotrait (HTMT) Criterion.

	Active Parental Mediation	Concentration	Enjoyment	Problematic TikTok Use	Time Distortion
Active Parental Mediation	–				
Concentration	0.225	–			
Enjoyment	0.279	0.755	–		
Problematic TikTok use	0.102	0.452	0.242	–	
Time Distortion	0.088	0.769	0.578	0.531	–

**Table 6 ijerph-20-02089-t006:** Direct Effect Hypotheses.

Hypothesis					Bootstrapped		
					CI	BC	
Variable Relationship	Path Coefficient Beta (β)	Standard Deviation (STDEV)	T Statistics (|O/STDEV|)	*p* Values	1% LL	99% UL	Decision
H1	0.714	0.022	32.676	0.000	0.656	0.760	Accept
H2	0.693	0.031	22.050	0.000	0.614	0.764	Accept
H3	−0.101	0.049	2.063	0.020	-0.227	0.012	Reject
H4	0.305	0.056	5.474	0.000	0.175	0.436	Accept
H5	0.371	0.051	7.239	0.000	0.255	0.483	Accept

Notes: Significant at *p* < 0.01.

**Table 7 ijerph-20-02089-t007:** Summary of Moderation Test Effects.

Hypothesis					Bootstrapped.		
					CI	BC	
Variable Relationship	Path Coefficient Beta (β)	Standard Deviation (STDEV)	T Statistics (|O/STDEV|)	*p* Values	1% LL	99% UL	Decision
H6	0.110	0.054	2.013	0.022	−0.027	0.235	Reject
H7	−0.145	0.061	2.363	0.009	−0.283	-0.031	Accept
H8	0.068	0.042	1.612	0.054	−0.035	0.164	Reject

Notes: Significant at *p* < 0.01.

**Table 8 ijerph-20-02089-t008:** Model results for R^2^ and Q^2^.

Dependent Variables	R^2^	Q^2^
Concentration	0.509	0.430
Time Distortion	0.481	0.388
Problematic TikTok Use	0.319	0.194

**Table 9 ijerph-20-02089-t009:** Summary of AVE (Average Variance Extracted) and R^2^.

Constructs	AVE	R^2^
Enjoyment	0.760	
Concentration	0.856	0.509
Time Distortion	0.818	0.481
Active Parental Mediation	0.641	
Problematic TikTok Use	0.641	0.319
Average	0.757	0.436

## Data Availability

Not applicable.

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
