# Peer review of "Flow Experience Is a Key Factor in the Likelihood of Adolescents’ Problematic TikTok Use: The Moderating Role of Active Parental Mediation"

_ijerph, 2023, doi:10.3390/ijerph20032089_

Round 1

Reviewer 1 Report

This study explored an interesting topic of problematic TikTok use. However, I have several concerns.

1.      The term “parental mediation” is quite difficult to understand, which was not clearly defined in this paper. It can easily be misunderstood by readers. I guess it is a translation problem.

2.      The scales used in the questionnaire need to be introduced with more details in the Methods section. The third paragraph in Methods is not enough and more details must be given especially your modifications.

3.      The measurement for “enjoyment” is problematic. The items in the scale (shown in Appendix A) are almost the same: “I think that using TikTok is enjoyable” “I think that using TikTok is pleasurable.”, “I think that watching TikTok video is interesting”. Besides, the variable called “enjoyment” sounds like a general emotion but you were testing situational (state) enjoyment while using TikTok.

4.      The term concentration has the same problem. Readers can easily regard “concentration” as another construct such as attentional concentration as a personal characteristic. The name of the research variables must be explained carefully.

5.      The goodness of fit indices of the final model need to be reported besides the R2 and Q2.

Reviewer 2 Report

The present work aims at examining the relationship between problematic TikTok use and flow experiences in terms of enjoyment, concentration, and time distortion. Moreover, the authors also investigated the moderating role of active parental mediation in the aforementioned relationship. In keeping with this, 633 Chinese adolescents (age range: 10-19 years old) participated in the study by completing an online survey. Results highlighted the existence of positive effects of concentration and time distortion on the problematic use of TikTok, and the significant mediating role of active parental control in the relationship between concentration and the risky behavior investigated. 

In general, I appreciated both the topic of this work and how it has been structured. The research topic refers to an emergent issue among adolescents that still need to be investigated. Particularly, considering that TikTok is one of the main social media used by adolescents, I repute that a study focused on this is able to implement the literature about it. 

The manuscript is smooth and easy to read without losing the relevant information by describing them in a very detailed way. From the introduction to the discussion, each section made referred accurately and precisely to the literature data. I considered it an added value. The authors provided a strong background to sustain their research hypothesis thus allowing the reader to follow the different phases of the project. From a methodological point of view, the authors clearly described each step made including research design, measurements, data collection, and common methods bias. This rigorous strategy has also been maintained for the description of data analysis and results. Moreover, the authors provided – both in the manuscript and appendix – explicative tables and figures able to synthesize the measurements used and results clearly and directly. Regarding the discussion section, I liked that the authors also explained non-significant results. Often, “negative results” are undervalued; however, I think that these last can strongly contribute to the development of future research, so much so that, it is important to consider them. I would try to better balance the discussion. More in detail, considering the long introduction and the rationale provided by the article, I’d elaborate more on the potential explanations of the study’s findings.  However, the authors extended the description of the implications of their study by partially supporting this request. In this regard, in line with what I expressed before, they minutely expanded the potential implications of their study by establishing a solid groundwork for future studies. Finally, with respect to the limitations, I suggest the authors specify that their study is not representative of the population investigated. I think that this specification does not reduce the value of their work, on the contrary, it gets better and it is in line with the rigorous approach used by the authors for the drafting of their manuscript.  

Other consideration and other minor revisions are reported below.

Abstract: The abstract is generally well written. It would useful to better rephrase and clarify from line 12 to line 14.

Line 15: If is possible, add also the mean age of the sample. You can report male’s percentage only, to gain some words to better explain the objective.

Introduction: 

Line 41: the reference [2] was expected to expand the association between TikTok interface and engaging content with flow experience. The article in question, never mention the flow theory or other cognitive “resemblance”. It instead starts from the “site attachment” and considers the personalization process and the functional dependence to social media.

Theoretical background and hypotheses development:

Line 78: a dot is needed to close the sentence.

Line 115: the reference [42] was expected to describe the flow experience as a positive predictor of problematic social media use. The article in question never mention “flow” or any other related construct but it rather considers the role of anxious and avoidant attachment in problematic social media use.

Methods:

It would be helpful to add in the appendix the graphical representation of the relationship existing between the variables under examination (Path Diagram).

Please, specify the sampling technique.

Discussion: Reporting “Findings” in the Discussion section risks to be redundant. That would be useful to leave the findings in the “results” section, in order to have more space to discuss the implication of the findings. 

Limitations and future research:

Express the representativeness of the statistical sample as a limit.

Conclusion: 

Line 456: delete the double dot.

Appendix A:

Please, specify the name of each scale.

Reviewer 3 Report

It is great to see some research investigating the phenomena of TikTok - I think the uniqueness of TikTok as a platform should be emphasised. It is not like any other social media. This article has a lot of potential.

There are a number of grammatical and noun-verb-tense errors throughout the article. Notably the phase in the title of 'Deeply flow experience' does not make sense to me. On line 443 of page 14 - 'conduct a cross-culture research' - should be 'conduct cross-cultural research' for instance.

Between 2.1 and 2.2 another section should be added defining what 'parental mediation' is, and that 'active' parental mediation is one of the types (in addition to section 2.3). Section 5.1.1 Implications focuses on the importance of parental mediation but the content covered in that 'third' implication focuses on restrictive mediation practices, rather than active mediation. See Padilla-Walker, L.M., Coyne, S.M., Fraser, A.M., Dyer, W.J., & Yorgason, J.B. (2012). Parents and adolescents growing up in the digital age: Latent growth curve analysis of proactive media monitoring. Journal of Adolescence, 35.

Most of the questions asked in the survey construct about active parental mediation are not about restrictive mediation practices. The use of the phrase 'active mediation control' in the abstract points to this tension I have identified. Control is not mediation.

Are you talking about parental mediation in a general sense, rather than a specific one?

Page 5, line 218, the authors state, 'the study sample is appropriate for this research'. Why? Please justify.

How were participants invited to participate and consent to their involvement in the study? How were they recruited?

More detail about how the survey construct was created is needed, especially the difference between the original and the modified items. Specifically, how did Nikken and Jansz's work influence the active parental mediation section of the survey? Should there not have been a pilot or validation of the survey?

What kind of 'experiments' (page 14, line 448) would be appropriate to use for future research?

It would be helpful to label the hypotheses in tables 7 and 8 with their numbers, i.e., H3 instead of FE-PTU .... I think section 5.1 could include more discussion about the acceptance of the 5/8 hypotheses that were tested.

Round 2

Reviewer 1 Report

Thanks for the modifications and responses. The paper is now ready for publication.